# Mediterranean-Type Diets as a Protective Factor for Asthma and Atopy

**DOI:** 10.3390/nu14091825

**Published:** 2022-04-27

**Authors:** Emilia Vassilopoulou, George V. Guibas, Nikolaos G. Papadopoulos

**Affiliations:** 1Department of Nutritional Sciences and Dietetics, International Hellenic University, 57400 Thessaloniki, Greece; 2Department of Allergy and Clinical Immunology, Royal Preston Hospital, Lancashire Teaching Hospitals NHS Foundation Trust, Preston PR2 9HT, UK; georgios.gkimpas@manchester.ac.uk; 3School of Biological Sciences, Medicine and Health, University of Manchester, Manchester M13 9PL, UK; nikolaos.papadopoulos@manchester.ac.uk; 4Allergy Department, 2nd Pediatric Clinic, National and Kapodistrian University of Athens, Thivon and Levadias 1, 11527 Athens, Greece

**Keywords:** Mediterranean diet, asthma, atopy, antioxidants, lipids, vitamin

## Abstract

We are currently riding the second wave of the allergy epidemic, which is ongoing in affluent societies, but now also affecting developing countries. This increase in the prevalence of atopy/asthma in the Western world has coincided with a rapid improvement in living conditions and radical changes in lifestyle, suggesting that this upward trend in allergic manifestations may be associated with cultural and environmental factors. Diet is a prominent environmental exposure that has undergone major changes, with a substantial increase in the consumption of processed foods, all across the globe. On this basis, the potential effects of dietary habits on atopy and asthma have been researched rigorously, but even with a considerable body of evidence, clear associations are far from established. Many factors converge to obscure the potential relationship, including methodological, pathophysiological and cultural differences. To date, the most commonly researched, and highly promising, candidate for exerting a protective effect is the so-called Mediterranean diet (MedDi). This dietary pattern has been the subject of investigation since the mid twentieth century, and the evidence regarding its beneficial health effects is overwhelming, although data on a correlation between MedDi and the incidence and severity of asthma and atopy are inconclusive. As the prevalence of asthma appears to be lower in some Mediterranean populations, it can be speculated that the MedDi dietary pattern could indeed have a place in a preventive strategy for asthma/atopy. This is a review of the current evidence of the associations between the constituents of the MedDi and asthma/atopy, with emphasis on the pathophysiological links between MedDi and disease outcomes and the research pitfalls and methodological caveats which may hinder identification of causality. MedDi, as a dietary pattern, rather than short-term supplementation or excessive focus on single nutrient effects, may be a rational option for preventive intervention against atopy and asthma.

## 1. Introduction

The prevalence of atopy and asthma has increased substantially in most developed countries [1,2,3], a trend potentially influenced by both genetic and environmental factors [4]. While our genetic profile is unlikely to have altered much in the last decades, our living conditions and habits have undergone major changes [5]. Environmental exposures are, therefore, considered to be the key culprit for the escalation [6]. As environmental factors may be amenable to intervention, their identification is important in efforts to curtail the current allergy epidemic. The observation that the rate of asthma has risen concomitantly with the degree of affluence [7] has kindled a search for specific cultural influences. It has been documented that culture shapes the asthma experience, its diagnosis and management, relevant research, and the politics determining funding [8].

As initially proposed by Burney [9] and later supported and extended by Seaton [10], changes in dietary habits could be partly responsible for the observed increase in atopic disease [11]. The rise in the prevalence of asthma in Western societies has coincided with marked modifications in the diet of these populations, leading to theories of a link between nutritional factors and asthma/atopy [12,13,14,15,16,17,18]. Among other factors, a link between a diet high in advanced glycation end-products (AGEs) and AGE-forming sugars and an increase in food allergy was proposed by Smith and colleagues [2]. It is thought that AGEs might function as a “false alarm” for the immune system against food antigens. In the pathogenesis of asthma/allergic airway inflammation, a receptor specific for the AGEs (RAGE) appears to be a critical participant [19].

A trend towards a lower prevalence of asthma in the Mediterranean region was identified in 1998 by the International Study of Asthma and Allergies in Childhood (ISAAC) [20], and epidemiological evidence has indicated that the Mediterranean diet (MedDi) might be protective against asthma [21,22,23,24].

Hence, a reasonable hypothesis is that adherence to the MedDi may modulate asthma pathogenesis, but even with a substantial body of research evidence, findings on such an association are far from conclusive. To account for the discrepant results, a variety of limitations, cultural, geographical, biological and methodological, must be taken into consideration. In this comprehensive narrative review, we present and discuss the currently available evidence on the relationship of the MedDi with asthma/atopy, we underscore the research pitfalls, and we propose ways forward, including possible interventions. We have opted to focus on the “contemporary Greek MedDi”, rather than the general archetype of the “Mediterranean diet of the previous century”, in an effort to circumvent one of the key research limitations, which is the diversity of the MedDi.

## 2. Obstacles to the Validation of Associations

### 2.1. Issue No. 1: What Is a “Mediterranean Diet”?

The basis of the MedDi model is the diet of the people of the island of Crete in the early 1950s [25]; it is characterized by a high plant/animal food ratio, and, compared with other populations, it is linked with a markedly low prevalence of chronic diseases, including cardiovascular disease (CVD), breast cancer, colorectal cancer, diabetes mellitus (DM), obesity, asthma, erectile dysfunction, depression and cognitive decline, and with a high life expectancy [26].

The typical MedDi pattern is composed of the following [25,27,28]: (1) daily consumption of refined cereals, and their products (bread, pasta, etc.), fruit (4–6 servings/day), vegetables (2–3 servings/day), olive oil (as the principal source of fat), wine (1–2 glasses/day) and dairy products (1–2 servings/day); (2) weekly consumption of fish, legumes, poultry, olives and nuts (4–6 servings/week), and (3) monthly consumption of red meat and meat products (4–5 servings/month). In sum, the MedDi is characterized by a high intake of plant foods such as fruits, vegetables, cereals, legumes, olive oil and nuts, a high to moderate intake of fish and seafood, a moderate to low intake of dairy products and wine, and only small quantities of red meat.

MedDi is, by most accounts, a generalization, if not a misnomer, as several variations are observed around the Mediterranean basin [29], which, unsurprisingly, reflects the diversity in religious, economic and social structures in these areas. For example, Muslims abstain from pork and wine, while Greek Orthodox populations usually avoid eating meat on Wednesdays and Fridays and during the 40-day fasting periods before major religious festivals. Differences stem also from the local availability of foodstuffs, which was a critical issue until some decades ago, as food transfer was arduous, and people had to rely on what they could produce and procure locally. In the 1960s, therefore, even different regions within the same Mediterranean country followed their own, distinct, dietary patterns [30].

Food transport is no longer a factor, but regional production continues to dictate, to some extent, the local cuisine. In parallel, food consumption patterns have changed during the last 50 years in most regions, including Crete [31], with adaptation to Westernized dietary patterns, leading to a poor MedDi quality index [32,33,34]. The cardinal feature of a Mediterranean-type diet, olive oil, however, still serves as the principal source of dietary fat in Crete, as in many Mediterranean regions, providing the precious monounsaturated fatty acids (MUFAs) and polyphenols [35,36].

### 2.2. Issue No. 2: Nutrients, Foods or Dietary Patterns?

Diet is a highly complex exposure variable. Traditional research approaches, focusing on individual nutrients, generate methodological pitfalls, as they may fail to take into account important nutrient interactions [37,38]. Perceived “healthy” foods contain numerous beneficial nutrients, necessitating stringent analytical adjustment to uncover their individual actions [39], but this approach would require sample sizes in excess of those reported in most published studies [40,41]. There are also conceptual issues; humans consume concurrently a variety of foods containing a constellation of nutrients, and the clinical relevance of the presumed effects of single nutrients is, therefore, questionable. It is reasonable to assume that any beneficial clinical effect is mediated through the combined action of several dietary agents [42,43], and the implication is that large-scale dietary manipulation, rather than single nutrient supplementation, would be a more promising approach for asthma-related research and possible disease prevention. In practice, the investigation of dietary patterns, rather than individual nutrients, is a paradigm that is gaining momentum [44]. To that end, the use of diet scores has been recommended, and various different MedDi scores have been constructed and used in research, including the Mediterranean diet score (MDS),the Mediterranean diet scale (MDScale) and the Mediterranean food pattern (MFP) [28,45,46]. All these indices show satisfactory performance in assessing adherence to the MedDi [47], and the MDScale and MedDi show correlation with olive oil and fiber constituents, while the MDScale shows correlation with waist-to-hip ratio and total energy intake [35].

It is of note that individual foods and constituents within the MedDi, specifically fish and olive oil appear to be particularly beneficial for asthma outcomes [48,49,50].

### 2.3. Issue No. 3: Observation versus Intervention

Findings on a possible correlation between diet and asthma are conflicting, but it appears that most of this disparity derives from a single methodological feature, namely, whether the studies are observational or interventional. A consistent theme in the epidemiological/cross-sectional literature is that of a beneficial effect of several dietary agents on asthma/atopy; this is in contrast with randomized clinical trials/supplementation studies, which yield inconclusive results. Several theories have been proposed to explain this discrepancy, such as the short duration of supplementation, or the requirement of an underlying deficiency for supplementation of the nutrient to show effects [51], both of which have been partly overthrown [12].

It has been suggested that positive observational evidence may stem from prenatal maternal nutrition, which could define the asthma risk of the offspring, and also serve as a model for the dietary habits of the child/grown adult [52,53]. In effect, observational studies could erroneously link the diet of the child/adult with favorable effects, when the defining factor may, in fact, be the maternal diet during pregnancy; this would also explain the failure of postnatal interventions. Although this theory has not been confirmed, it is plausible that the maternal prenatal diet may modify the future asthma risk of the child via epigenetic “programming” of the fetal lung and immune system [54]. Interest in the role of modifiable nutritional factors specific to both the prenatal and the early postnatal life is increasing, as during this time the immune system is particularly vulnerable to exogenous influences. A variety of perinatal dietary factors, including maternal diet during pregnancy, duration of breastfeeding, use of special milk formulas, timing of the introduction of complementary foods, and prenatal and early life supplementation with vitamins and probiotics/prebiotics, have all been addressed as potential targets for the prevention of asthma [55].

Breastfeeding is a sensitive period, and knowledge of its effects has, to date, been gained observationally [56,57,58]. The results related to the possible protection gained through breastfeeding against allergies and asthma have been inconsistent [59,60]. Current evidence suggests a protective role of exclusive breastfeeding in atopic dermatitis (AD), related to atopic heredity [61].

Reduced intake of omega 3 (n-3) polyunsaturated fatty acids (PUFAs) may be a contributing factor to the increasing prevalence of wheezing disorders [62,63], and a higher ratio of n-6/n-3 PUFAs in the maternal diet, and maternal asthma, increase the risk of wheeze/asthma in the offspring [63]. The effect of n-3 PUFA supplementation in pregnant women on the risk of persistent wheeze and asthma in their offspring has been assessed. Supplementation with n-3 long-chain (LC)PUFA in the third trimester of pregnancy was shown to reduce the absolute risk of persistent wheeze and lower respiratory tract infection (LRTI) in infancy [64,65], and the risk of asthma by the age of 5 years [66], and later in life [67].

## 3. Evaluation of Dietary Constituents

Two major research hypotheses have been proposed to explain the link between dietary constituents and asthma: the lipid hypothesis and the antioxidant hypothesis. Both hypotheses are based on foods that are hallmarks of the MedDi. A third one, the anti-inflammatory hypothesis emerges to merge and corroborate the other two.

### 3.1. The Lipid Hypothesis

In 1997, Black and Sharpe [68] originally suggested that changes in the intake of fatty acids (FAs), in both type and quantity, has contributed to the rise of asthma and atopy in the West. FAs are categorized as saturated or unsaturated, depending on the presence of double carbon bonds, as shown in Figure 1. Unsaturated FAs are classified into MUFAs, such as oleic acid, and PUFAs, which are further divided into subgroups, the n-6 and the n-3 PUFAs, based on the position of the double carbon bonds.

The n-6 PUFAs have proinflammatory properties [69]; for example, linoleic acid (LA), which is a common n-6 PUFA and the principal FA in the US diet [70], is converted into arachidonic acid (ARA), which is further metabolized by cyclooxygenase (COX) and lipoxygenase into 2-series prostanoids and 4-series leukotrienes [71]. The proinflammatory and immunomodulatory properties of these agents, their promotion of a Th2 phenotype, and their association with bronchoconstriction, are well established [72,73]. A causal link between increased intake of n-6PUFAs and a high incidence of allergic disease has been suggested, which is supported by biologically plausible mechanisms, related to the role of eicosanoid mediators produced from the n-6 PUFA ARA [74]. Conversely, the n-3PUFAs, exemplified by a linolenic acid (ALA), exert an anti-inflammatory action by restricting the metabolism of ARA [75].

As shown in Figure 1, this proposed beneficial effect of the ALA catabolism products, eicosapentaenoic acid (EPA, n-3) and docosahexaenoic acid (DHA, n-3) stems from the competitive inhibition of LA(n-6) catabolism; EPA and DHA, naturally occurring in fish oil [76], also downregulate COX-2 gene expression and activity and suppress neutrophil function [77].

It was therefore postulated that the rising prevalence of atopy in affluent societies was preceded by reduced consumption of oily fish, which contain n-3 PUFAs, and increased intake of margarine and vegetable oils rich in n-6 PUFAs, which were favored by public health measures aimed at decreasing cholesterol levels by the replacement of butter, which contains saturated FAs [78]. It is likely that such dietary modifications affect asthma mechanisms in several ways, and hence, the controversy over the lipid hypothesis continues [79]. For example, FAs may influence the Th cells and the synthesis of Th1/Th2-associated cytokines, which playa basic role in the cell membrane regulating protein function, membrane fluidity and gene expression [80,81].

All PUFAs are necessary for normal epidermal structure and function, and a reduction in the levels of both n3 and n6 PUFAs, brought about by an inherent abnormality of D6-desaturase, has been suggested as paving the way for the presentation of AD [82,83]. The consumption of oily fish rich in the n-3 PUFAs EPA and DHA [84] is an integral constituent of the lipid hypothesis. The original MedDi score did not include high fish intake, but as fish is also a key constituent of the contemporary MedDi, this parameter was added in a later modification [85]; the people of Crete have been reported to consume up to 30-fold more fish than their US peers [86]. Whether there is a meaningful difference between oily and non-oily fish is under debate [87,88], and at least one study has reported similar protective effects of maternal prenatal intake of non-oily fish and intake of both non-oily and oily fish, on the development of asthma/atopy in the offspring [89]; hence, an atopy-modifying effect of non-oily fish cannot be dismissed.

Both observational and intervention studies support a protective effect of prenatal maternal fish intake on asthma/atopy in the offspring (Table 1), but in the epidemiological approach, a variety of factors appear to modify this effect diversely, and occasionally contradictorily. In one US case-control study, monthly oily fish intake during pregnancy was associated with a reduced asthma risk in childhood, but only in children born to asthmatic mothers [87]. A high maternal plasma n-6/n-3 PUFA ratio in the second trimester of pregnancy was associated with current wheeze, current asthma and diagnosed asthma in 1019 children at the age of 4 to 6 years; male sex and maternal asthma increased the risk of wheeze and asthma [63]. Conversely, in a cross-sectional study from Italy, weekly prenatal fish consumption protected from skin prick test (SPT)-evidenced atopy, but only in children born to non-allergic mothers [90]. In a study from Mexico, weekly fish consumption during pregnancy safe guarded the offspring of mothers both with and without a history of allergy from AD, SPT-evidenced atopy and wheeze; adjustment for breastfeeding nullified the wheeze-related effect [91]. In a racially diverse cohort of 1131 pregnant women in the US, 67% were African-American and 42% had a history of atopic disease; 17% of their children had AD, and a higher level of n-6 PUFAs in the second trimester of pregnancy was associated with AD in the children of women with atopy [43].

In another study, high maternal plasma levels of EPA, DHA and total n-3 FAs at 34 weeks of gestation were associated with a reduced risk of non-atopic persistent/late wheeze at the age of 6 years, in 865 children born full term [95].

Lower rates of AD were associated with weekly maternal prenatal fish consumption in one longitudinal study [162], and with high fish consumption during the last 4 weeks of pregnancy in another prospective study [162]. Maternal fish oil supplementation from the 20th week of gestation until delivery was associated with a lower risk of positive SPT to egg and lower severity of AD (94), and with higher levels of n-3 PUFA in neonatal erythrocytes and lower levels of plasma interleukin-13 (IL-13) in the offspring [96]. In a population-based study of 533 Danish pregnant women, fish oil supplementation protected the offspring from asthma, up to the age of 16 years [94], and in a Swedish study, such supplementation was associated with a lower rate of food allergy and AD in one-year-old infants [163]. Other prenatal intervention studies suggest that maternal n3 supplementation could be irrelevant [164], or even detrimental [165], for the risk of atopy in the offspring [165], but these findings were preliminary and/or the studies were not either powered or designed to detect clinical outcomes.

Although the evidence on prenatal exposure may appear convincing, certain concerns have been expressed. The first is the proposed D6-desaturase theory: In one study, atopy was positively associated with lower levels of n-6 products in the cord blood [118]; this is consistent with the D6-desaturase hypothesis, which ascribes a protective role against AD to all PUFAs (i.e., both n-3 and n-6), in contrast with the presumed asthma-facilitating effect of n-6 PUFAs. According to this theory, asthma and AD must be viewed differently, with n-3PUFAs being protective for both entities and n-6 PUFAs only for AD. Another concern pertains to the transfer of PUFAs from the mother to the fetus, and stems from reports that the association of cord blood PUFA levels with childhood wheeze/atopy disappeared on control for multiple comparisons [166] and that oily fish supplementation in pregnancy modifies neonatal immune responses, but may not affect markers of infant atopy assessed at 6 months of age [167]. This poses a problem, as, for the lipid hypothesis to have validity in the prenatal context, a link between LC-PUFAs transferred via cord blood and later atopy would be expected. In the LISAplus cohort study, the levels of n-6 LC-PUFA and n-3 LC-PUFA in cord blood serum and the n-6/n-3 ratio showed no significant association with eczema, asthma, hay fever/allergic rhinitis, or aeroallergen sensitivity [116]. In another cohort study, maternal shellfish consumption during pregnancy was associated with an increased risk of wheezing and eczema in the offspring [168], while fatty fish consumption was associated with a higher risk of eczema only, and total fish or lean fish consumption was not associated with either wheezing or eczema [169]. In a population-based cohort study of 4260 mother-child pairs, maternal plasma FA patterns during mid-pregnancy showed no association with the child’s asthma and allergy outcome at 10 years of age. In the KOALA Birth Cohort Study, eczema in early childhood (6–7 years) was associated with higher prenatal exposure to n-6 than n-3 PUFAs [117], confirming the association between PUFA levels in maternal blood and offspring allergy demonstrated in earlier studies [170].

The n-3 PUFA content of breast milk is strongly related to maternal fish intake [171,172] and, therefore, the possibility of the passage of PUFAs to the infant through breastfeeding should be considered. Lower levels of n-3 PUFAs in mature milk have been related to the development of atopy in breast-fed babies [173,174], predominantly in those with mothers with extensive allergic disease [175]. Considerable temporal variability is observed in the PUFA content of breast milk, independent of dietary habits; the levels have been shown to differ between mothers of term and preterm babies [176,177], between the late gestation (colostrum) and later lactation periods [178,179], and even over a single day [180]. In one study, high n-3 PUFA levels in colostrum were identified as a risk factor for later atopy in the infant [181]. This evidence leads to speculation that a temporal variability in the PUFA composition of breast milk may also characterize cord blood; indeed, in one study, considerable individual variation in cord blood PUFA levels was reported [181], which would disallow conclusions made on the basis of one-time PUFA measurements in cord blood. 

Reports have been published of maternal fish oil supplementation attenuating cord blood lipid peroxidation [182], and being associated with higher n3 levels in the erythrocytes of the neonates [183] and lower mRNA levels of Th2 cytokines in the cord blood [184], all suggesting a strong maternal/offspring PUFA link.

In the postnatal/adult observational setting, although findings are inconclusive, reports of a protective effect of fish intake predominate, and numerous epidemiological studies report a beneficial effect on asthma and atopy of n-3 PUFA intake from fish [21,69,97,98,99,100,101,102,103]. In the interventional setting, the evidence is far less conclusive and, in general, points towards a lack of effect. Supplementation with EPA, n-3 showed no clinical benefit in subjects with AD [104,105], pediatric asthma [106], mild [107], uncontrolled [107] or severe asthma [108] and in patients with hay fever [109,110]; it even had a detrimental effect on asthma in aspirin-intolerant patients [114,115]. 

In the Childhood Asthma Prevention Study (CAPS), one of the largest randomized controlled trials (RCTs) undertaken to date, in spite of the promising early findings [111], in 616 infants the maternal antenatal n-3 PUFA supplementation failed to confer protection from atopy/asthma at 5 years of age [74]. Conversely, some studies report favorable effects of n-3 PUFA supplementation [94], with the Copenhagen Prospective Studies on Asthma in Childhood 2010 (COPSAC2010), involving 700 mother-child dyads demonstrating a 31% reduction in the risk of asthma during the first 5 years of life [66]. Similarly, systematic reviews and meta-analyses have provided conflicting results on the protective effect of n-3 PUFA/fish oil supplementation on asthma and/or AD [112,113].

To explain the discrepancies between observational and intervention studies in the postnatal setting, the differential effects of n-6 PUFAs on AD and asthma may be considered. Interventional studies and a Cochrane review indicated a favorable effect of postnatal n-6 PUFA supplementation on AD [119,120]. This result was not confirmed for AD and asthma, by several studies that reported a proinflammatory effect of the intake of margarine, which is rich in n-6 PUFAs, but not a common item in the Greek diet [121,122]. In addition, temporal/phenotypic differences should be considered; for example, postnatal fish oil supplementation was associated with wheeze reduction at 18 months, but not at 3 years of age [121]. This implies phenotype-specific differences [43,185], as also suggested by a differential response to lipid supplementation in individual patients with asthma [186]. Other factors which may lead to diverse effects of PUFA supplementation may include dosage, cytokine imbalance [187] and different IgE levels according to the age of the subjects [96,188]. The efficiency of conversion of precursor FAs by an FA desaturase (FADS) variant might play a significant role, as proposed by Talaei and colleagues, in the Avon Longitudinal study of Parents and Children, where they showed that children with a higher intake of EPA and DHA from fish had a lower risk of incident asthma up to adolescence [189].

PUFA intake in the typical Greek diet is characterized by a favorable n-6/n-3 ratio of 2:1, compared with 15:1 in Western and Northern Europe and 74:1 in the US [86]. This ratio is affected by other factors in addition to fish consumption. The traditional Greek diet includes wild green leafy plants, which are a rich source of n-3 PUFAs [190] and are exemplified by purslane, a commonly consumed plant rich in locked nucleic acid (LNA) (n-3) [190,191]. In addition, the meat, milk and cheese derived from animals raised free-range in the Greek countryside contain high levels of n-3 PUFAs, as the animals graze on leafy green plants, rather than being fed grain [192].

Overall, the current evidence, although diverse, may reveal a pattern. Prenatal and infantile, via early breastfeeding, exposure to fish oil/n-3 PUFAs appears to have a beneficial effect on asthma outcomes. Postnatal observational studies in children/adults also furnish convincing evidence. As postnatal interventions typically fail, it is possible that the favorable epidemiological findings may reflect, not current fish intake, but rather a programming of the immune system in fetal or early infantile life; considerable evidence supports the existence of a critical early “time window”, during which future immunological behavior is determined [193,194,195,196].

In the CHILD cohort study, in a selected subgroup of 1109 mother-infant dyads, 184 infants (17%) were found to be sensitized to one or more food allergens, and 160 (14%) presented AD. Human milk content of PUFAs and their ratios were associated with sex-specific infant atopic conditions; in female infants, a higher ARA/DHA ratio may reduce the risk of food sensitization and AD [171]. Evidence from another survey suggests that postnatal PUFA supplementation may be protective against childhood asthma and allergy [197]. The outcomes observed so far, however, cannot justify a recommendation for fish oil supplementation for asthma prevention purposes [74,110], although in our opinion, they provide sufficient grounds to suggest that high habitual fish intake (as in the contemporary MedDi), may partly contribute to a decrease in asthma prevalence. 

The MedDi is also very low in trans FAs, the main constituent of industrially hydrogenated vegetable fats, used widely in fast food [198] (Figure 1). Trans FAs have been documented to be associated with asthma and atopy in several studies [123,124,125,126], but they are not commonly consumed around the Mediterranean basin [199,200]. The Greek version of the MedDi, akin to the Cretan diet of the 1960s, is low in saturated FAs (SFAs) (Figure 1). Although the percentage of overall energy intake from SFA consumption in Greece has admittedly increased from roughly 8% in the 1960s, it is unlikely to have risen considerably, as Greeks have a cultural distaste for animal fats [86]; therefore, although up to 20% was reported in the early 1990s [201] the 10–15% reported at the turn of the century is a more likely estimate [199]. Hence, in the MedDi, SFA consumption is much lower than in Western diets [202], an obvious advantage, as high SFA intake is documented to induce oxidative stress [203] and to be positively associated with bronchial hyperresponsiveness [127], asthma [128] and atopy [129]. Finally, a hallmark of a Greek MedDi is a high intake of MUFA, in the form of oleic acid (n-9), the main FA of olive oil [24,86,92] which is inversely associated with asthma [79,204] (Figure 1). Olive oil was always an integral feature of the Cretan diet and remains the principal constituent of the Greek diet [205], with up to 22% of the overall energy intake originating from MUFAs [206]. Olive oil competes neither with the catabolism of n-3 PUFAs nor with their incorporation into the cell membrane, hence supporting their functions [93]. Furthermore, it is rich in vitamin E and other antioxidants, including tocopherols, oleuropein, hydroxytyrosol and other polyphenols [207]. Thus, olive oil may be protective against asthma/atopy via both of the proposed mechanisms, the lipid hypothesis and the antioxidant hypothesis.

### 3.2. The Antioxidant Hypothesis

The antioxidant hypothesis was proposed in 1994 by Seaton and colleagues [10], who suggested that a Westernized diet, progressively deficient in antioxidants, could be held accountable for the rising prevalence of atopy. Accumulating evidence indicates the possible involvement of oxidative stress in the pathophysiology of inflammatory disorders such as asthma and allergic rhinitis [44,208,209].

The lungs are susceptible to oxidative injury because of their high oxygen environment, large surface area and rich blood supply [210]; hence the respiratory system has evolved elaborate antioxidant defenses, including three key antioxidant enzymes, superoxide dismutase and glutathione peroxidase and catalase, and numerous non-enzymatic antioxidant compounds, including glutathione, vitamins C and E, α-tocopherol, lycopene, β-carotene, and others. Disruption in the redox balance may favor asthma induction [211].

Maternal consumption of a MedDi rich in fruits, vegetables, fish and vitamin D-containing foods has been shown to exert possible benefit against the development of allergies and asthma [21], or at least to protect small airway function, in childhood [212]. Fruit and vegetables commonly consumed by Mediterranean populations [24], particularly those produced locally, such as oranges, cherries, grapes and tomatoes [213,214], are rich in antioxidants, such as glutathione, vitamin C, vitamin E, vitamin A and provitamin A carotenoids (α-carotene, β-carotene, β- cryptoxanthin, lutein/zeaxanthin and lycopene), and polyphenols (primarily flavonoids) [44,214]. Numerous observational studies have suggested a protective effect of high postnatal (childhood/adult) consumption of such foods against asthma/allergic rhinitis [50,215] and atopy/AD [132,213].

Interpretation of these findings should not overlook the effect of specific micronutrients on the outcomes investigated. For instance, low dietary vitamin C intake, and low levels of serum such as corbate have been associated with current wheeze [133], asthma and reduced ventilatory function in numerous epidemiological studies, possibly confirming that vitamin C is a major bronchial antioxidant [134,135]. McEvoy and colleagues in a randomized controlled trial (RCT) of vitamin C supplementation to pregnant smokers showed better pulmonary function at 3 months of age in the offspring of mothers taking vitamin C [216].

Supplementation with vitamin C, however, of even up to 16 weeks, has generally yielded poor results [136], as also concluded by a Cochrane review [137]. Vitamin C supplementation may benefit exercise-induced bronchoconstriction [134], and treatment with high doses of intravenous vitamin C was suggested to control allergy symptoms [138]. The compounded evidence points towards a protective effect of a longstanding habitual diet rich in vitamin C, rather than short-or medium-term supplementation and/or intervention in already established asthma [43,133].

The case for vitamin E is similar; vitamin E is the principal defense against oxidant-induced membrane injury [217]; its dietary sources are olive oil, olives, nuts and avocado [218], which are key constituents of the MedDi. Vitamin E, in contrast to vitamin C, exerts additional non-antioxidant immune-regulating action in the form of the modulation of IL-4 gene expression, production of eicosanoid and IgE, neutrophil migration and allergen-induced monocyte proliferation [219,220].

Intervention studies in the postnatal setting have produced conflicting findings [141,142,143]. Considerable epidemiological evidence suggests a protective role against asthma for vitamin E [143] although there are also reports of no benefit [144], possibly indicative of the opposing regulatory effects of the tocopherol isoforms of vitamin E [145]. Favorable reports have been produced in the prenatal observational setting, with some studies showing a protective effect of high maternal prenatal consumption on asthma/AD outcomes in the offspring [130]. This may reflect the immune modulating action of vitamin E during a critical period for immune system ontogeny. As with vitamin C, habitual dietary intake of vitamin E may protect against asthma, but this may be due to interaction between closely related nutrients; for example, concurrent vitamin C and E supplementation was observed to protect against bronchoconstriction in one cross-sectional study in preschool children [139], but firm conclusions could not be drawn on the effectiveness of vitamin C and E on either asthma control or exercise-induced bronchoconstriction, in a systematic review by Wilkinson and colleagues [140].

Another important antioxidant vitamin contained in fruit and vegetables is vitamin A, which is a group composed of retinol and over 600 carotenoids (β-carotene, β-cryptoxanthin, lutein-zeaxanthin, lycopene, etc.) [221]. A beneficial association between dietary carotenoids and bronchial function and asthma outcome has been demonstrated in several postnatal epidemiological studies [12], and high carotenoid dietary intake has been proposed to protect pulmonary function and metabolic health in obese asthmatic children [146,147]. Other studies, however, have reported no protective effect [148] and, overall, current evidence on the protective effect of vitamin A is inconclusive.

Vitamin D3 (cholecalciferol) is in a different category, as its potential effects are dependent, not on antioxidant action, but on other activities, including regulation of gene expression and chemokine secretion, reversal of steroid resistance and a variety of immunomodulatory functions, including modulation of acyl carrier protein (APC) and Treg ontogeny, inhibition of antigen-specific T cell activation, and others [222]. Dietary vitamin D is derived from milk and fish intake, but a major source is the cutaneous production under exposure to sunlight [223], which is abundant around the Mediterranean throughout the year, whereas northern populations are not adequately exposed to sunlight. Vitamin D deficiency is common among populations from different geographical areas, and is related to gene polymorphisms, sunlight exposure and dietary intake [224]. Vitamin D supplements and the fortification of foods, such as bread, cereals, and dairy products, are proposed to ensure adequate intake [224]. Epidemiological findings regarding high prenatal maternal levels of vitamin D are conflicting, supporting in general, but not consistently [12,13], a favorable effect on bronchial outcomes in the offspring, but not necessarily asthma [149,150]. Recently, Jensen and colleagues reported an association of low serum levels of total 25-hydroxyvitamin-D[(25(OH)D] in pregnant women with asthma with a greater risk of adverse respiratory outcomes in the offspring at 12 months [151]. Wolk and colleagues, based on a combined analysis of two metanalyses, concluded that vitamin D supplementation during pregnancy significantly reduced the risk of asthma/recurrent wheeze in the offspring, especially among women with 25(OH)D levels ≥30 ng/mL at randomization, where the risk was almost halved [225]. Results from two RCTs did not support this conclusion, however, as prenatal supplementation with vitamin D did not reduce the incidence of asthma or recurrent wheeze among children at 6 years of age who were considered at risk [154,226].

A protective effect of vitamin D against respiratory tract infections has been proposed to explain these conflicting findings [152,153]. Early life interventions indicated no protective effect of high dose vitamin D against the risk of persistent wheeze and asthma in children, except when the mother had asthma [154,227], in which case vitamin D appeared to attenuate the risk conferred by maternal asthma on childhood asthma and recurrent wheeze [155]. Postnatal observational studies show favorable effects of vitamin D, and postnatal intervention trials support a protective bronchial and anti-viral effect [157,158,159], but a clear beneficial effect remains to be confirmed, especially with regard to non-respiratory outcomes (Table 1). 

Studies have demonstrated that airway epithelial cells, lung fibroblasts, airway smooth muscle cells, and immune cells (T and B cells, macrophages, monocytes and dendritic cells) are equipped with vitamin D receptors and high levels of the enzyme, 1α-hydroxylase [228,229]. In children with sufficient serum levels of 25(OH)D at baseline, a MedDi enriched with fatty fish was associated with enhanced pulmonary function [160]. Studies have documented a high rate of vitamin D deficiency in children with asthma (≥40%), and a dose-response relationship between serum vitamin D and spirometry parameters FEV1, FVC, and FEV1/FVC, but not FeNO, which suggests a potentially important role for vitamin D in respiratory health. Lower levels of vitamin D are associated with poorer asthma control, and it is possible that normalization of vitamin D status could lower the asthma burden in children with asthma, and reduce the associated costs [161].

Flavonoids are polyphenolic plant metabolites found in fruits, vegetables, nuts, seeds and wine [230], hence constituting a common nutrient of the Greek diet. They exert a wide array of biological functions, including antioxidant, anti-inflammatory and antiallergic [231]. Luteolin and apigenin, which are contained in celery and parsley in large amounts, are strong inhibitors of IL-4, IL-13, tumor necrosis factor α (TNF-α) and of cysteine-leucotriene synthesis, phospholipase A2 action and basophil CD40 ligand expression [232,233]. Although flavonoids effectively counter AD and asthma in experimental rodent models [234,235], epidemiological evidence is scarce and conflicting, with some studies reporting an unequivocal protective bronchial effect, but others failing to show any benefit [236].

A higher intake of dietary fiber, mostly insoluble fiber and fiber from cereals, has been shown to mediate an anti-inflammatory effect and to be associated with fewer asthma symptoms and better asthma control [237]. Certain types of dietary fibers, known as microbiota-accessible carbohydrates, function as a valuable feeding resource for the microbiome ensuring its stability and surveillance [238,239]. Dietary fibers are degraded by carbohydrate active enzymes, encoded by specific bacterial strains in the microbiome [240,241]. Different types of dietary fibers supply different subsets of microbes [242], and it is hypothesized that they modulate the diversity of the microbiome, its metabolic function and the immunological outcomes [242]. 

Soy isoflavone treatment was reported to reduce the number of severe asthma exacerbations in patients with the high plasminogen activator inhibitor-1(PAI-1)-producing genotype of asthma, and PAI-1 polymorphisms were suggested as a genetic biomarker for soy isoflavone-responsive asthma [243]. Disappointingly, soy isoflavone supplements in children aged 12 years or older and adults with poorly controlled asthma taking a controller medication, did not improve lung function significantly [244].

Red wine produced from “black” grapes is consumed in moderation around the Mediterranean basin. Flavonoids, which are present in red wine, have been shown to exert antioxidant, anti-inflammatory, anticancer, and immunomodulatory activities [245]. Black grape skin extract contains non-flavonoid and flavonoid polyphenols [246], and especially 3,49,5-trihydroxystilbene (resveratrol), a polyphenol which can inhibit the formation of lipoxygenase products and downregulate inducible nitric oxide (NO) synthase expression and bronchial nitrite production [247]. Red wine intake has been associated with lower asthma prevalence and severity, and reduced numbers of exacerbations [248], although the evidence is scant and inconclusive. Overall, the current evidence of a possible protective effect of flavonoids on human asthma is modest, at best.

Finally, the trace element selenium (Se) has been the focus of several studies, because of its prominent antioxidant properties, specifically its incorporation into glutathione peroxidase, a key constituent of the lung antioxidant defenses [249]. The major dietary sources of Se are several plant and animal foods, but the content varies according to the location [250]. Fish and seafood consumed in Mediterranean countries are rich in Se, and the population intake in these countries is quite high [251]. From a prenatal perspective, a significant association was observed between cord blood Se and the frequency of fish consumption in women from four Mediterranean countries; the Se content in breast milk from Greek women was also shown to be quite high [243]. Low serum levels of Se are common in patients with asthma [252,253], but without firm conclusions on the potential for reverse causality. In one intervention study in 24 patients, 14 weeks of Se supplementation led to clinical improvement of asthma, but did not alter objective markers of the disease [254]. In the observational setting, including a large multicenter case-control study, no effect of Se was shown on asthma outcomes [255], and overall, the current evidence on the effects of Se is inconsistent.

In conclusion, attempting to postulate on the effect of antioxidant intake on the prevention of atopy/asthma, several factors must be considered, including the diverse findings in each setting for different antioxidant agents and their interrelationships.

The current evidence, overall, builds a strong case in favor of a protective effect against asthma/atopy of a diet based on antioxidant-rich fruit and vegetables, especially in the form of habitually high, life-long consumption, exemplified by the traditional MedDi [256]. 

### 3.3. The Anti-Inflammatory Hypothesis

An emerging scenario used to explain the inter-relation of diet and the development of lifestyle-related chronic diseases, including atopy and asthma, is the anti-inflammatory hypothesis [257,258]. Diet may be a route to either inducing or halting systemic inflammation in patients with asthma through modulation of the innate system [259]. A high intake of saturated fat directly activates the toll-like receptor 4 (TLR4), which leads to a Nf-kappa B-driven inflammatory cascade. In obese subjects with asthma, the metabolically active adipose tissue releases proinflammatory mediators, including IL-6, TNF-α and C-reactive protein (CRP), and adipokines such as leptin, which are central to the innate immune pathways [259]. Conversely, an adequate intake of n-3 PUFAs [260], vitamins E, C, β-carotene [134] and magnesium [261], fiber and moderate alcohol intake [262] favor protection from systemic inflammation. 

In 2013, Shivappa and colleagues, developed a literature-derived population-based index, the Dietary Inflammation Index (DII), which measures the potential impact of a diet on the inflammatory status of an individual. The overall score depends on the whole diet and not on certain nutrients or foods [263]. A high DII score has been associated with increased systemic inflammation and lower lung function in subjects with asthma, confirming the hypothesis that a proinflammatory diet leads to a worsening of asthma symptoms in adults [264] and in children with atopy [43]. To date, no study has investigated in parallel the DII score and the adherence to MedDi. The anti-inflammatory effects of MedDi in patients with atopy/asthma have mainly been indicated by the clinical outcomes [132,265] but direct examination, with measurement of specific inflammatory markers has been extremely limited; Douros and colleagues reported an association between higher adherence to MedDi in children with asthma and lower levels of IL-4, IL-33 and IL-17 [266].

## 4. Conclusions: Mediterranean-Type Dietary Pattern?

In conclusion, the findings from intervention studies which disallow recommendations for supplementation seem unconvincing, and the effect of short-term excessive dietary intake is also questionable. The relevant guidelines on asthma management need to be tailored to take into consideration the heterogeneity of real-life settings [267]. It appears that long-standing dietary patterns, alongside other pertinent lifestyle factors, rather than short-term supplementation with specific constituents, may hold the key for a protective effect against asthma/atopy [268]. Such patterns guarantee appropriate nutrient interplay and discourage excessive focus on single nutrient effects, which is the first main research pitfall; they guarantee a temporal continuum and ensure that the diet will be consistent during potentially critical “time windows”, whether pregnancy, infancy or early childhood, which is the second key research pitfall; finally, they guarantee life-long even distribution of reasonable amounts of nutrients, rather than short-term, and potentially detrimental, periodic excessive intake, which is the third key research pitfall. Within this conceptual framework, a habitual MedDi emerges as a rational option for preventive intervention against atopy and asthma.

## Figures and Tables

**Figure 1 nutrients-14-01825-f001:**
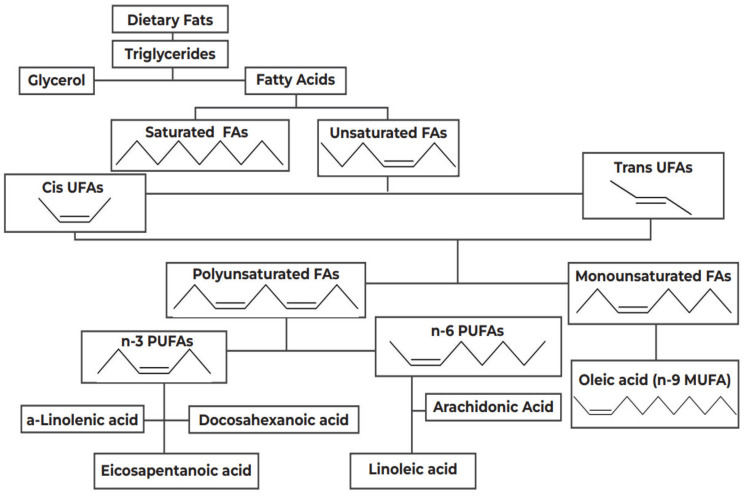
Categorization of dietary fats. FA: Fatty Acids; PUFAs: Polyunsaturated Fatty Acids; MUFA: Monounsaturated Fatty Acids.

**Table 1 nutrients-14-01825-t001:** Synopsis of the effect of monounsaturated fatty acids (MUFAs) and n3 polyunsaturated fatty acids (PUFAs), and vitamins A, C and E on asthma and atopy outcomes, and the effect of Vitamin D on bronchial and atopy outcomes.

Type of Lipids	Asthma/Atopy Outcomes	Reference
MUFAs	• Reduced risk of asthma/atopy	[70,92]
	• Cell membranes support-reduced AD	[93]
n-3 PUFAs (fish oils)	Prenatal	
	• reduced risk of wheeze/asthma	[62,63,66,67,87,94]
	• respiratory infections	[65]
	• atopy	[90,91,95]
	• AD	[96]
	Postnatal	
	observational: protection on asthma, atopy	[26,69,97,98,99,100,101,102,103]
	interventional: inconclusive on	
	• AD	[104,105]
	• asthma/wheeze	[66,106,107,108,109,110,111,112,113]
	• worsening on asthma in aspirin-intolerant patients	[114,115]
n-6 PUFAs	Prenatal	
	no effect or increased risk of	
	• AD	[43,116,117]
	• atopy	[118]
	Postnatal	
	• contradictory results on AD and asthma	[119,120,121,122]
Trans-fat	Increased risk of asthma/atopy	[123,124,125,126]
Saturated FAs	Increased	
	• bronchial hyperresponsiveness	[127]
	• asthma	[128]
	• atopy	[129]
Vitamins C, E, A (fruits and vegetables)	Prenatal	
	• observational: conflicting evidence on asthma/atopy	[130]
	• interventional: lack of evidence	
	Postnatal	
	• observational: generally favorable on asthma/wheeze, atopy	[43,50,131,132,133,134,135,136,137,138,139,140]
	• interventional: conflicting evidence, nutrient specific	[12,141,142,143,144,145,146,147,148]
Vitamin D	Prenatal	
	• observational: generally favorable on bronchial outcomes	[12,13,149,150,151,152,153]
	• interventional: favorable when mother has asthma	[154,155,156]
	Postnatal	
	• observational: favorable on asthma/atopy	[157,158,159]
	• interventional: favorable on asthma/atopy	[160,161]

## Data Availability

Not applicable.

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
