# Peer review of "Mediterranean-Type Diets as a Protective Factor for Asthma and Atopy"

_nutrients, 2022, doi:10.3390/nu14091825_

Round 1

Reviewer 1 Report

This article systematically reviewed Mediterranean-type diets and asthma and atopy. It detailed underlying mechanisms of how nutrients rich in Mediterranean diet may play roles in protecting against asthma or atopy. There are few comments should be addressed:

1) Please summarize the findings or conclusions in the abstract. 

2) The article explained two hypotheses that linked dietary components and asthma: the lipid hypothesis and the antioxidant hypothesis. It is important to also include anti-inflammatory hypothesis. It has been reported that anti-inflammatory micronutrients or a diet with lower DII (dietary inflammatory index) is associated with decreased risk of asthma or atopy. Mediterranean diet is rich in anti-inflammatory micronutrients. 

3) Please expand discussion on how diet rich in fibers and low in energy intake (decrease in overweight or obese) may protect against asthma or atopy

4) Please update Table 1. It is rather difficult to read/understand

5) There are long paragraphs of how different types of lipids may associate with asthma or atopy. Would it be possible to summarize the study findings in a Table by type of lipids (i.e. MUFAs, PUFAs, n-3 PUFAs, n-6 PUFAs, trans-fat)  

Author Response

There are few comments should be addressed:

  1. Please summarize the findings or conclusions in the abstract. 

Thank you for your comment. A relevant summary of the conclusions has been added in the abstract

  1. The article explained two hypotheses that linked dietary components and asthma: the lipid hypothesis and the antioxidant hypothesis. It is important to also include anti-inflammatory hypothesis. It has been reported that anti-inflammatory micronutrients or a diet with lower DII (dietary inflammatory index) is associated with decreased risk of asthma or atopy. Mediterranean diet is rich in anti-inflammatory micronutrients. 

The anti-inflammatory effect of MedDi/ specific micronutrients is discussed thoroughly throughout the text.

The DII has not been extensively studied for atopy/asthma and MedDi. Therefore, we incorporated a relevantly short part under the antioxidant hypothesis section, also to avoid repetitions (lines 1340-1365).

  1. Please expand discussion on how diet rich in fibers and low in energy intake (decrease in overweight or obese) may protect against asthma or atopy

We expanded the relevant topics in lines 1162-1171, 1376-1378 and 948-953, 1362-1368 respectively.

  1. Please update Table 1. It is rather difficult to read/understand
  2. There are long paragraphs of how different types of lipids may associate with asthma or atopy. Would it be possible to summarize the study findings in a Table by type of lipids (i.e. MUFAs, PUFAs, n-3 PUFAs, n-6 PUFAs, trans-fat)  

Following comments 4 and 5, we have made substantial modification in Table 1, in terms of content and design. We hope it is according to your expectations.

Reviewer 2 Report

This is an excellent review that starts off with a fascinating and very nicely written history and description of the Mediterranean diet and then delves into a very ambitious review of the literature with regard to specific components of the diet. While comprehensively researched and extensively cited, it is extremely challenging to summarize and synthesize such a complex literature, and I believe there are a few important papers/points that go unmentioned. Otherwise, my comments are all very minor.

1) Table 1: While the manuscript does a nice job of describing areas where data are conflicting, this table does not appear to have comprehensive enough citations to support its conclusions. For example, in the interventional studies sections, there are several clinical trials that go uncited (see studies referenced in the recent review PMID: 32524677 DOI: 10.1111/pai.13303). It may be more feasible to focus this table on a smaller topic, such as one on evidence regarding the Mediterranean diet pattern itself without delving into its component nutrients.

2) Lines 315-317: In discussion of role of PUFA in aspirin-intolerant asthma, please review and reference: Schneider TR, Johns CB, Palumbo ML, Murphy KC, Cahill KN, Laidlaw TM. Dietary Fatty Acid Modification for the Treatment of Aspirin-Exacerbated Respiratory Disease: A Prospective Pilot Trial. J Allergy Clin Immunol Pract. 2018 May-Jun;6(3):825-831. doi: 10.1016/j.jaip.2017.10.011. Epub 2017 Nov 10. PMID: 29133219; PMCID: PMC5945343.

3) PUFA section: In describing factors that lead to study heterogeneity, it may be worthwhile to mention FADS genotype variants as potential effect modifiers of PUFA-atopy associations, e.g.:

Talaei M, Sdona E, Calder PC, Jones LR, Emmett PM, Granell R, Bergström A, Melén E, Shaheen SO. Intake of n-3 polyunsaturated fatty acids in childhood, FADS genotype and incident asthma. Eur Respir J. 2021 Sep 2;58(3):2003633. doi: 10.1183/13993003.03633-2020. PMID: 33509958; PMCID: PMC8411098.

4) Vitamin C section: Would be worthy to include a brief discussion of the evidence for prenatal vitamin D in asthma prevention in offspring of smoking mothers, e.g.:

McEvoy CT, Shorey-Kendrick LE, Milner K, Schilling D, Tiller C, Vuylsteke B, Scherman A, Jackson K, Haas DM, Harris J, Park BS, Vu A, Kraemer DF, Gonzales D, Bunten C, Spindel ER, Morris CD, Tepper RS. Vitamin C to Pregnant Smokers Persistently Improves Infant Airway Function to 12 Months of Age: A Randomised Trial. Eur Respir J. 2020 Jul 2:1902208. doi: 10.1183/13993003.02208-2019. Epub ahead of print. PMID: 32616589; PMCID: PMC8029653.

5) Vitamin D section: The weight of evidence supports antenatal vitamin D supplementation in prevention of offspring wheeze, but not necessarily persisting asthma. The following papers would be useful to cite in this discussion:

- Wolsk HM, Chawes BL, Litonjua AA, Hollis BW, Waage J, Stokholm J, Bønnelykke K, Bisgaard H, Weiss ST. Prenatal vitamin D supplementation reduces risk of asthma/recurrent wheeze in early childhood: A combined analysis of two randomized controlled trials. PLoS One. 2017 Oct 27;12(10):e0186657. doi: 10.1371/journal.pone.0186657. PMID: 29077711; PMCID: PMC5659607.

 - Litonjua AA, Carey VJ, Laranjo N, Stubbs BJ, Mirzakhani H, O'Connor GT, Sandel M, Beigelman A, Bacharier LB, Zeiger RS, Schatz M, Hollis BW, Weiss ST. Six-Year Follow-up of a Trial of Antenatal Vitamin D for Asthma Reduction. N Engl J Med. 2020 Feb 6;382(6):525-533. doi: 10.1056/NEJMoa1906137. PMID: 32023372; PMCID: PMC7444088.

 - Brustad N, Eliasen AU, Stokholm J, Bønnelykke K, Bisgaard H, Chawes BL. High-Dose Vitamin D Supplementation During Pregnancy and Asthma in Offspring at the Age of 6 Years. JAMA. 2019 Mar 12;321(10):1003-1005. doi: 10.1001/jama.2019.0052. PMID: 30860552; PMCID: PMC6439670.

6) Lines 72-75: perhaps this could come later in section 2.1 as it isn’t clear this early in the ms what is meant by “contemporary Greek” vs “MedDi of the previous century.”

7) Citation 64: please cite the study itself (PMID: 28029926 DOI: 10.1056/NEJMoa1503734) rather than an editorial.

8) Citation 189: This Cochrane review has been withdrawn and replaced with more recent ones as described here: https://www.cochranelibrary.com/cdsr/doi/10.1002/14651858.CD000993.pub4/full. Please update the reference.

9) Line 447: Please provide a reference for the 2014 systematic review by Wilkinson.

10) There seem to be spaces missing between words throughout the manuscript.

11) Figure 1 is low resolution/blurry.

12) Line 125: do you mean “promising” instead of “prolific”?

Author Response

This is an excellent review that starts off with a fascinating and very nicely written history and description of the Mediterranean diet and then delves into a very ambitious review of the literature with regard to specific components of the diet. While comprehensively researched and extensively cited, it is extremely challenging to summarize and synthesize such a complex literature, and I believe there are a few important papers/points that go unmentioned.

Thank you very much for your overall comment. We hope that the amendments in the manuscript, based to your comments, are according to your expectations.

Otherwise, my comments are all very minor.

1) Table 1: While the manuscript does a nice job of describing areas where data are conflicting, this table does not appear to have comprehensive enough citations to support its conclusions. For example, in the interventional studies sections, there are several clinical trials that go uncited (see studies referenced in the recent review PMID: 32524677 DOI: 10.1111/pai.13303). It may be more feasible to focus this table on a smaller topic, such as one on evidence regarding the Mediterranean diet pattern itself without delving into its component nutrients.

Following your comment, we have made substantial modification in Table 1, in terms of content and design. We hope it is according to your expectations.

2) Lines 315-317: In discussion of role of PUFA in aspirin-intolerant asthma, please review and reference: Schneider TR, Johns CB, Palumbo ML, Murphy KC, Cahill KN, Laidlaw TM. Dietary Fatty Acid Modification for the Treatment of Aspirin-Exacerbated Respiratory Disease: A Prospective Pilot Trial. J Allergy Clin Immunol Pract. 2018 May-Jun;6(3):825-831. doi: 10.1016/j.jaip.2017.10.011. Epub 2017 Nov 10. PMID: 29133219; PMCID: PMC5945343.

 We corrected this omission. Thank you for noticing it.

3) PUFA section: In describing factors that lead to study heterogeneity, it may be worthwhile to mention FADS genotype variants as potential effect modifiers of PUFA-atopy associations, e.g.:

Talaei M, Sdona E, Calder PC, Jones LR, Emmett PM, Granell R, Bergström A, Melén E, Shaheen SO. Intake of n-3 polyunsaturated fatty acids in childhood, FADS genotype and incident asthma. Eur Respir J. 2021 Sep 2;58(3):2003633. doi: 10.1183/13993003.03633-2020. PMID: 33509958; PMCID: PMC8411098.

 Thank you for this recommendation. We included a relevant part in lines 364-369.

4) Vitamin C section: Would be worthy to include a brief discussion of the evidence for prenatal vitamin C in asthma prevention in offspring of smoking mothers, e.g.:

McEvoy CT, Shorey-Kendrick LE, Milner K, Schilling D, Tiller C, Vuylsteke B, Scherman A, Jackson K, Haas DM, Harris J, Park BS, Vu A, Kraemer DF, Gonzales D, Bunten C, Spindel ER, Morris CD, Tepper RS. Vitamin C to Pregnant Smokers Persistently Improves Infant Airway Function to 12 Months of Age: A Randomised Trial. Eur Respir J. 2020 Jul 2:1902208. doi: 10.1183/13993003.02208-2019. Epub ahead of print. PMID: 32616589; PMCID: PMC8029653.

 Thank you for the recommendation. We included the reference in our manuscript (lines 443-446)

5) Vitamin D section: The weight of evidence supports antenatal vitamin D supplementation in prevention of offspring wheeze, but not necessarily persisting asthma. The following papers would be useful to cite in this discussion:

- Wolsk HM, Chawes BL, Litonjua AA, Hollis BW, Waage J, Stokholm J, Bønnelykke K, Bisgaard H, Weiss ST. Prenatal vitamin D supplementation reduces risk of asthma/recurrent wheeze in early childhood: A combined analysis of two randomized controlled trials. PLoS One. 2017 Oct 27;12(10):e0186657. doi: 10.1371/journal.pone.0186657. PMID: 29077711; PMCID: PMC5659607.

 - Litonjua AA, Carey VJ, Laranjo N, Stubbs BJ, Mirzakhani H, O'Connor GT, Sandel M, Beigelman A, Bacharier LB, Zeiger RS, Schatz M, Hollis BW, Weiss ST. Six-Year Follow-up of a Trial of Antenatal Vitamin D for Asthma Reduction. N Engl J Med. 2020 Feb 6;382(6):525-533. doi: 10.1056/NEJMoa1906137. PMID: 32023372; PMCID: PMC7444088.

 - Brustad N, Eliasen AU, Stokholm J, Bønnelykke K, Bisgaard H, Chawes BL. High-Dose Vitamin D Supplementation During Pregnancy and Asthma in Offspring at the Age of 6 Years. JAMA. 2019 Mar 12;321(10):1003-1005. doi: 10.1001/jama.2019.0052. PMID: 30860552; PMCID: PMC6439670.

 Thank you for your significant recommendations. All citations are incorporated in the document (lines 500-507).

6) Lines 72-75: perhaps this could come later in section 2.1 as it isn’t clear this early in the ms what is meant by “contemporary Greek” vs “MedDi of the previous century.”

 We decided not to move this downwards, as it is part of the aim of the study. We do hope you will agree with this decision.

7) Citation 64: please cite the study itself (PMID: 28029926 DOI: 10.1056/NEJMoa1503734) rather than an editorial.

 We replaced the reference. Thank you

8) Citation 189: This Cochrane review has been withdrawn and replaced with more recent ones as described here: https://www.cochranelibrary.com/cdsr/doi/10.1002/14651858.CD000993.pub4/full. Please update the reference.

 Corrected. Thank you

9) Line 447: Please provide a reference for the 2014 systematic review by Wilkinson.

The citation is added. Thank you

10) There seem to be spaces missing between words throughout the manuscript.

 We revised the document accordingly and corrected throughout accordingly.

11) Figure 1 is low resolution/blurry.

Figure 1 is re-designed to increase the resolution. Thank you

12) Line 125: do you mean “promising” instead of “prolific”?

Thank you for noticing this mistake. We corrected it.

Reviewer 3 Report

Thank you for the opportunity to review the manuscript titled, “Mediterranean-type diets as a protective factor for asthma and atopy.” In this manuscript, the authors aimed to review the current literature reporting on the associations between the components of the Mediterranean diet and asthma/atopy. In particular, the authors considered pathophysiological links between the Mediterranean diet and disease outcomes, and research and methodological gaps that limit identification of causality. This was an exceptionally well-written and interesting review, which I very much enjoyed reading. I have only some minor comments.

1.    A small concern, but one that nonetheless warrants mention. In the field of allergy, the word “component” already carries at least two distinct meanings, neither of which captures the authors’ use of this word in the aim. Please consider replacing “components” [of a Mediterranean diet] with another word.
2.    Figure 1 is a bit blurry. Please adjust for sharpness.
3.    Figure 1. Please add abbreviations in the text boxes that correspond with those in the main text. This will facilitate the reader’s ability to follow the figure while reading the text.
4.    Please add some language, and ideally, a flow chart, as to how the included articles were selected.
5.    The manuscript contains some minor language and syntax errors. While these error do not impede comprehension, they are nonetheless distracting and should be corrected.

Author Response

Thank you for the opportunity to review the manuscript titled, “Mediterranean-type diets as a protective factor for asthma and atopy.” In this manuscript, the authors aimed to review the current literature reporting on the associations between the components of the Mediterranean diet and asthma/atopy. In particular, the authors considered pathophysiological links between the Mediterranean diet and disease outcomes, and research and methodological gaps that limit identification of causality. This was an exceptionally well-written and interesting review, which I very much enjoyed reading. I have only some minor comments.

  1. A small concern, but one that nonetheless warrants mention. In the field of allergy, the word “component” already carries at least two distinct meanings, neither of which captures the authors’ use of this word in the aim. Please consider replacing “components” [of a Mediterranean diet] with another word.

We highly appreciate your comment. Although component is commonly used in literature in this context, for instance https://doi.org/10.1016/B978-0-12-820593-8.00020-3; https://doi.org/10.1186/1741-7015-12-100; doi: 10.1016/j.amjmed.2014.10.014; doi:10.1002/mnfr.201801095, we have now used constituent instead.

  1. Figure 1 is a bit blurry. Please adjust for sharpness.

We have made proper amendments to the figure, and we hope the result is satisfactory

  1.    Figure 1. Please add abbreviations in the text boxes that correspond with those in the main text. This will facilitate the reader’s ability to follow the figure while reading the text.

Thank you for your useful comment. Proper changes have been applied in the figure.

  1.    Please add some language, and ideally, a flow chart, as to how the included articles were selected.

A relevant text is included in lines 75-85.

  1.    The manuscript contains some minor language and syntax errors. While these error do not impede comprehension, they are nonetheless distracting and should be corrected.

English language editing has been conducted by a native English speaker